# Inflammatory Bowel Disease Patients’ Acceptance for Switching from Intravenous Infliximab or Vedolizumab to Subcutaneous Formulation: The Nancy Experience

**DOI:** 10.3390/jcm11247296

**Published:** 2022-12-08

**Authors:** Clotilde Remy, Bénédicte Caron, Celia Gouynou, Vincent Haghnejad, Elodie Jeanbert, Patrick Netter, Silvio Danese, Laurent Peyrin-Biroulet

**Affiliations:** 1Department of Gastroenterology, University Hospital of Nancy, F-54000 Nancy, France; 2NGERE (Nutrition-Génétique et Exposition aux Risques Environnementaux), U1256 INSERM, Université de Lorraine, F-54000 Nancy, France; 3Unit of Methodology, Data Management and Statistic, Nancy University Hospital, F-54000 Vandoeuvre-lès-Nancy, France; 4Ingénierie Moléculaire et Ingénierie Articulaire (IMoPA), UMR-7365 CNRS, Faculté de Médecine, University Hospital of Nancy, University of Lorraine, F-54000 Vandoeuvre-lès-Nancy, France; 5Gastroenterology and Endoscopy, IRCCS Ospedale San Raffaele, University Vita-Salute San Raffaele, 20132 Milano, Italy

**Keywords:** inflammatory bowel disease, subcutaneous, infliximab, vedolizumab

## Abstract

Background: Subcutaneous infliximab and vedolizumab formulations have been developed for maintenance therapy in inflammatory bowel disease. The objective of this study was to explore the inflammatory bowel disease patient’s acceptance for switching from intravenous infliximab or vedolizumab to subcutaneous, as well as to describe the causes of refusal or, conversely, the factors associated with acceptance. Methods: Patients were prospectively recruited between June 2021 and March 2022 during their infusion of infliximab or vedolizumab in the Medical Day Hospital of Nancy University Hospital. Adult patients with an established diagnosis of inflammatory bowel disease in clinical remission were eligible for inclusion in this study if they had been treated with intravenous infliximab or vedolizumab for at least six months. Results: One hundred and thirty patients were included in this study. Thirty-six patients (27.7%) received vedolizumab and ninety-four patients (72.3%) received infliximab. Median duration of treatment at inclusion was 7.0 years [3.0–11.0]. In this cohort, 77.7% of patients accepted the switch from intravenous infliximab or vedolizumab to subcutaneous. The main reasons for patient’s refusal for switching from intravenous to subcutaneous formulation were fear of loss of efficacy, a more spaced-out medical follow-up, increased frequency of administration, and self-administered injection. A short duration of treatment was associated with a high switch acceptance rate (odd ratio (OR) (95% confidence interval (CI)) = 0.9 (0.8–0.9); *p* = 0.0002). Conclusion: A large majority of the patients included accepted the switch of their treatment with infliximab or vedolizumab from the intravenous form to the subcutaneous form. This study identified one predictor influencing the acceptance rate in inflammatory bowel disease population: short treatment duration. Subcutaneous infliximab and vedolizumab hold potential for greater patient flexibility by self-administration, reducing travel and hospital attendance for infusion.

## 1. Introduction

Inflammatory bowel disease (IBD) is a heterogeneous group of chronic inflammatory disorders: the main phenotypes comprise ulcerative colitis (UC) and Crohn’s disease (CD) [1,2]. Both are characterized by histologic chronic inflammation, periods of clinical relapse and remission, use of medication, risk of surgery, and impaired quality of life [1,2].

Guidelines recommend tumor necrosis factor (TNF) inhibitors, including infliximab for the treatment of moderately to severely active UC and CD that have not responded to conventional therapy [3,4]. As reference product patents have expired, infliximab biosimilars have been developed and had increasing uptake for the treatment of IBD [5].

A subcutaneous (SC) CT-P13 formulation has been developed for maintenance, providing opportunities for CT-P13 self-administration [6,7]. In 2020, SC CT-P13 received regulatory approval from the European Medicines Agency (EMA) for all indication previously approved for CT-P13, including IBD [8].

Vedolizumab is an anti α4β7-integrin monoclonal antibody that selectively blocks lymphocyte trafficking to the gut. It is approved as an intravenous (IV) formulation to treat moderately to severely active UC and CD [3,4]. The efficacy and safety of vedolizumab 300 mg intravenous as both induction and maintenance therapy is well established [9,10]. In 2020, a new formulation for SC of vedolizumab was developed to offer this option to patients who may prefer the convenience of SC therapy for maintenance [11].

SC biologics may benefit patients with IBD, including through improved ease of use, increasing convenience, and the reduced requirement for medical visits and associated travel [12]. SC biologics also offer potential benefits for healthcare systems by optimizing medical resources and reducing associated costs [13,14,15]. On the other hand, acceptability of the treatment needs to be investigated. In a French study, they highlighted that the main limiting factors of a treatment are the interval between two doses and the patient’s previous medication [16]. Recently, in a retrospective study on adult patients receiving standard dosing infliximab or vedolizumab maintenance therapy, the switch uptake rates were 58%, with 90% of patients eligible to switch [17]. Awaiting replication of these data is necessary due to cultural and healthcare system differences.

The objective of this study was to investigate the IBD patient’s acceptance for switching from an IV infliximab or IV vedolizumab to its SC formulation, and the factors associated with acceptance.

## 2. Methods

### 2.1. Study Design 

Patients were prospectively recruited in this observational study between June 2021 to March 2022 during their infusion of infliximab or vedolizumab in the Nancy University Hospital. All patients eligible for inclusion were offered the possibility of switching from IV CT-P13 to SC CT-P13 (120 mg every 2 weeks) or IV vedolizumab to SC vedolizumab (108 mg every 2 weeks). In case of non-acceptance of the switch from the IV form to the SC form of their treatment, they were asked the reason for the non-acceptance. Every patient performed their first self-injection with our advanced practice nurse at the hospital.

### 2.2. Study Population

Adult patients with an established diagnosis of IBD (CD or UC) may be included in this study if they were currently treated and received intravenous infliximab or vedolizumab since at least six months. No limits were set about prior or concomitant therapies. All patients with IBD, including those with perianal CD, were approached. At inclusion, patients were in clinical remission, which was defined by a Harvey–Bradshaw Index less than 4 for CD and a partial Mayo score less than 3 for UC.

### 2.3. Data Collection

At inclusion, sociodemographic data (age, gender) and clinical data (age at diagnosis, disease duration, regular tobacco use, type of IBD, location, phenotype, previous IBD surgery, IBD treatment at inclusion, median duration of current treatment at inclusion, and median therapy trough level) were collected. Data on disease activity were also collected (HBI or Mayo score, C-reactive protein, and fecal calprotectin). Collected data were strictly limited to the purpose of this study. Data were provided by patients themselves during their medical consultation or extracted from their electronic medical record. The management of the electronic dataset was realized under conditions guaranteeing the confidentiality of data and the anonymity of patients.

### 2.4. Statistical Analysis

The distribution of the responses against different questions for the entire dataset was analyzed.

Categorical variables were expressed as percentages, while continuous variables were described by median and Interquartile range [IQR] or min/max values, depending on their distribution. A logistic regression model was used to determine which factors were associated with the acceptance of the switch to a subcutaneous formulation of their current treatment. Firstly, a bivariable logistic regression model was used and variables with a *p*-value < 0.2 were candidates for the multivariable logistic regression model. Associations were described by odds ratios (OR) and 95% confidence intervals (CI). The significance threshold was set at 5%. These statistical analyses were performed using SAS version 9.4 software (SAS Institute).

## 3. Results

### 3.1. Patient Characteristics

A total of 130 patients were included in the study. Baseline characteristics are shown in Table 1. There were 66 women (50.8%) and 64 men (49.2%). The median [IQR] age at inclusion was 45.5 [17.0] years, and there were 27 active smokers (20.8%). Among this cohort, 44 patients had UC (33.8%), and 86 patients has CD (66.2%). The median age at IBD diagnosis was 26.0 [18.0] years, and median disease duration was 14.0 years [11.0].

Among the patients with CD, the disease was limited to the ileum in 26 cases (30.6%), and to the ileum and colon in 37 (43.5%). Thirty-one patients (36.0%) with CD had a history of perianal disease. Among the patients with UC, the majority had left-sided colitis (40.9%) or pancolitis (52.3%). Thirty-seven patients (28.5%) had previous surgery for IBD.

At inclusion, 36 patients (27.7%) received vedolizumab and 94 patients (72.3%) infliximab. The majority of patients with infliximab had CD (75.5%), and 58.3% of patients with vedolizumab had UC (Table 2). Median duration treatment at inclusion was 7.0 years [8.0]. Seventeen patients (13.1%) had a concomitant treatment at inclusion: 5-ASA (7.7%), azathioprine (2.3%), and methotrexate (3.1%).

Among this cohort, 76 patients (58.5%) needed dose escalation: 59 patients (77.6%) treated with infliximab and 17 patients (22.4%) with vedolizumab. For patients treated with infliximab, 14 (10.6%) received their treatment every 4 weeks, 16 (17.0%) every 6 weeks, 60 every 8 weeks (63.8%) and 8 (8.5%) every 10 or 12 weeks. For patients treated with vedolizumab, 12 received their treatment every 4 weeks (33.3%), 4 every 6 weeks (11.1%), 19 every 8 weeks (52.8%) and one every 12 weeks (2.8%) (Table 2).

At inclusion, median vedolizumab trough level was 15.0 ug/mL [17.3], and median infliximab trough level was 7 ug/mL [6.5].

### 3.2. IBD Patient’s Acceptance for Switching from an IV Infliximab or IV Vedolizumab to Its SC Formulation

In this cohort, 77.7% of patients agreed to switch from an IV treatment to its SC formulation (Table 3). It represents 83.3% patients receiving vedolizumab and 75.5% receiving infliximab. Among patients needing dose escalation, 55 (79.7%) agreed to switch from an IV treatment to its SC formulation. In the global cohort, patients who agreed to switch had a median treatment duration of 7.0 years [7.0], and patients who refused the switch had a median treatment duration of 12.0 years [6.0] (1.9 versus 3.4 years respectively in patients treated with vedolizumab) (Table 4). In patients treated with infliximab, the median duration of treatment was higher in patient who agreed to switch (5.4 vs. 4.5 years respectively) (Table 5).

In the global cohort, the patient’s non-acceptance reasons of the switch were a fear of loss of efficacy for ten patients (34.5%), a more spaced-out medical follow-up for ten patients (34.5%), increased frequency of administration for three patients (10.3%), and self-administered injection for four patients (13.8%) (Table 3). In the infliximab subgroup, the patient’s non-acceptance reasons of the switch were a negative impact on medical follow-up for nine patients (39.1%), a fear of loss of efficacy for seven patients (30.4%), self-administered injection for three patients (13.0%), and increased frequency of administration for two patients (8.7%). In the vedolizumab subgroup, the patient’s non-acceptance reasons of the switch were a fear of loss of efficacy for three patients (50.0%), a negative impact on medical for one patient (16.7%), increased frequency of administration for one patient (16.7%), follow-up and self-administered injection for one patient (16.7%).

### 3.3. Factors Associated with Patient’s Acceptance for Switching from IV to SC Formulation

In the multivariate analysis, a short duration of treatment was significantly associated with a higher switch acceptance rate (OR = 0.9, CI 95% 0.8–0.9, *p* = 0.0002) (Table 6). In contrast, the type of IBD or the current treatment (infliximab or vedolizumab) were not associated with patient’s acceptance rate for switching from IV to SC formulation.

## 4. Discussion

The advent of SC formulations of infliximab and vedolizumab offers patients and IBD units the opportunity to switch patients established on IBD therapy while being maintained on the same biologic. The availability of infliximab and vedolizumab in a SC formulation provides a useful alternative for patients requiring these treatments, with the possibility of self-administration at home. The purpose of the study was to investigate the IBD patient’s acceptance for switching from an IV infliximab or IV vedolizumab to its SC formulation, and to identify predictors associated with patient’s acceptance for switching.

An acceptability of 77.7% in the 130 patients with IBD included in this study and followed at Nancy University Hospital was identified. Recently, a retrospective study on adult patients receiving standard dosing infliximab or vedolizumab maintenance therapy evaluated the uptake and rationale for choosing to switch from IV infusions to SC injections. Switch uptake rates were 58%, with 90% of patients eligible to switch [17]. In our study, all patients with IBD in clinical remission treated with infliximab or vedolizumab at standard dosing or intensified dosing regimens were included. In the study of Burdge et al., patients on intensified dosing regimens were excluded [17].

In our study, one of the main reasons for non-acceptance of the switch was the fear of loss of efficacy of the treatment. Based on previous studies, SC infliximab efficacy is non-inferior to the originator’s IFX both in de novo treatment and after switch [7,18]. SC vedolizumab is an effective maintenance therapy in patients who responded to two infusions of vedolizumab IV induction therapy and after switch [19,20,21,22,23]. One of the reasons for not accepting the switch was the fear of a negative impact on follow-up due to the interval between medical consultations. After months of injection in a medical center, patients trust the medical staff and maybe feel that their disease is more under control if the treatment process is left to professionals [14]. Patient education is essential in this case and should always be proposed to patients in order to involve them in the cure of their chronic disease. Haghnejad et al. explored the impact of a gastroenterologist’s interview on patients with IBD for switching the infliximab bio-originator to its biosimilar [24]. They showed that the provision of organized information to the patient is a contributive way to enhance patient’s acceptance of biosimilars in IBD [24]. They were 1.47 times more likely to agree to the switch if the interview modified the patient’s opinion on biosimilars [24]. These data highlight the importance of a therapeutic education program to better inform patients and involve them in decision-making when, for example, changing the administration mode of a medication.

In our study, the frequency of administration of the SC formulation of infliximab or vedolizumab was a reason for not switching. Buisson et al. showed that acceptability is highly impacted by the rhythm of administration [16]. In patients with IBD, SC treatment with long intervals between two injections (≥8 weeks) seems to be one of the most accepted modalities [16].

Our study is the first to report that a short duration of treatment was independently associated with a high switch acceptance rate. The probability to accept the switch from IV to SC was reduced by 10% for each year of IV treatment. Age, sex, diagnosis, drug, and line of treatment were not predictors for acceptance of switching. In the study of Burdge et al., duration of treatment was not a predictor for willingness to switch [17]. The main strength of our study is the large number of patients. Participation to our study required a clinical remission and at least six months of IV therapy. All patients with IBD, including those needing dose escalation, were approached. However, there was no follow-up of patients after the switch to the subcutaneous formulation.

In conclusion, more than three quarters of the patients included accepted the switch of their treatment with infliximab or vedolizumab from the IV form to the SC form. This study identified one predictor influencing the acceptance rate in the IBD population: short treatment duration. SC infliximab and vedolizumab hold the potential for greater patient flexibility via self-administration, which reduces travel and hospital attendance for infusion. The implementation of a switch program with the opportunity to discuss this option with an IBD nurse or physician should improve this acceptance rate.

## Figures and Tables

**Table 1 jcm-11-07296-t001:** Patients characteristics.

Characteristics			Total (n = 130)	Crohn’s Disease (n = 86)	Ulcerative Colitis (n = 44)
Male gender, n (%)			64 (49.2)	38 (44.2)	26 (59.1)
Age median [IQR]			45.5 [17.0]	44.0 [16.0]	40.0 [20.0]
Age at diagnosis, median [IQR]			26.0 [18.0]	24.5 [14.8]	26.0 [22.0]
Disease duration at inclusion, median [IQR]			14.0 [11.0]	15.0 [9.5]	10.5 [11.5]
UC location, n (%)					
	Proctitis				3 (6.8)
	Left sided colitis				18 (40.9)
	Pancolitis				23 (52.3)
CD phenotype, n (%)					
	Inflammatory			62 (72.1)	
	Stricturing			11 (12.8)	
	Penetrating			13 (15.1)	
CD location, n (%)					
		Ileal		26 (30.6)	
		Colonic		17 (20.0)	
		Ileocolonic		37 (43.5)	
		Upper gastrointestinal		5 (5.9)	
Perianal disease, n (%)				31 (36.0)	
Previous surgery for IBD, n (%)			37 (28.5)	32 (37.2)	5 (11)
Current treatment, n (%)					
		Vedolizumab	36 (27.7)	15 (17.4)	21 (47.7)
		Infliximab	94 (72.3)	71 (82.6)	23 (52.3)
		Biosimilar	57 (60.6)	42 (59.1)	15 (65.2)
		Remicade^®^	37 (39.4)	29 (40.9)	8 (34.8)
Treatment dose escalation (%)			76 (58.5)	53 (61.6)	23 (52.3)
	Infliximab		59 (77.6)	44 (51.2)	15 (34.1)
	Vedolizumab		17 (22.4)	9 (10.5)	8 (18.2)
Median vedolizumab trough level (ug/mL), median [IQR]			15.0 [17.3]	20.0 [19.1]	14.3 [11.8]
Median infliximab trough level (ug/mL), median [IQR]			7.0 [6.5]	7.1 [7.6]	6.5 [5.0]
Concomitant treatment, n (%)					
		None	113 (86.9)	79 (91.9)	34 (77.3)
		5-aminosalicylates	10 (7.7)	2 (2.3)	8 (18.2)
		Azathioprine	3 (2.3)	2 (2.3)	1 (2.3)
		Methotrexate	4 (3.1)	3 (3.5)	1 (2.3)
Median duration of treatment (year), median [IQR]			7.0 [8.0]	8.0 [7.0]	4.0 [5.5]
Administration frequency					
	Every 4 weeks		22 (16.9)	14 (16.3)	8 (18.2)
	Every 6 weeks		20 (15.4)	14 (16.3)	6 (13.6)
	Every 8 weeks		79 (60.8)	51 (59.3)	28 (63.6)
	Every 10 to 12 weeks		9 (6.9)	7 (8.1)	2 (4.5)

IBD, inflammatory bowel disease; UC, ulcerative colitis; CD, Crohn’s disease; IQR, interquartile range; n, number.

**Table 2 jcm-11-07296-t002:** Comparison of patient characteristics according to treatment.

		Total		Vedolizumab		Infliximab		
		N= 130		N = 36 (27.7%)		N = 94 (72.3%)		
								*p* **
Gender, n (%)								0.4993
	Male	64	(49.2)	16	(44.4)	48	(51.1)	
	Female	66	(50.8)	20	(55.6)	46	(48.9)	
IBD, n (%)								0.0003
	UC	44	(33,8)	21	(58.3)	23	(24.5)	
	CD	86	(66.2)	15	(41.7)	71	(75.5)	
Previous surgery for IBD, n (%)								0.1585
	No	93	(71.5)	29	(80.6)	64	(68.1)	
	Yes	37	(28.5)	7	(19.4)	30	(31.9)	
Treatment dose escalation, n (%)								0.2223
	No	61	(46.9)	20	(55.6)	41	(43.6)	
	Yes	69	(53.1)	16	(44.4)	53	(56.4)	
Concomitant treatment, n (%)								0.8553
	None	113	(86.9)	33	(91.7)	80	(85.1)	
	5 ASA	10	(7.7)	2	(5.6)	8	(8.5)	
	Azathioprine	3	(2.3)	0	(0.0)	3	(3.2)	
	Methotrexate	4	(3.1)	1	(2.8)	3	(3.2)	
Median duration of treatment (year)			5.3		2.6		5.5	<0.0001
Administration frequency, n (%)								0.0051
	Every 12 weeks	1	(0.8)	1	(2.8)	0	(0.0)	
	Every 10 weeks	8	(6.2)	0	(0.0)	8	(8.5)	
	Every 8 weeks	79	(60.8)	19	(52.8)	60	(63.8)	
	Every 6 weeks	20	(15.4)	4	(11.1)	16	(17.0)	
	Every 4 weeks	22	(16.9)	12	(33.3)	10	(10.6)	

** Chi-2 test or Fisher’s exact test for qualitative variables, Wilcoxon test for quantitative variables. CD, Crohn’s disease; IBD, inflammatory bowel disease; UC, Ulcerative Colitis.

**Table 3 jcm-11-07296-t003:** Reasons for patient’s refusal for switching from IV to SC formulation.

Reasons for Patient’s Refusal for Switching from IV to SC Formulation	Total	IFX	VDZ
Fear of loss of efficacy, n (%)	10 (34.5)	7 (30.4)	3 (50.0)
Impact on medical follow-up	10 (34.5)	9 (39.1)	1 (16.7)
Increased frequency of administration	3 (10.3)	2 (8.7)	1 (16.7)
Self-administered injection	4 (13.8)	3 (13.0)	1 (16.7)
Other	2 (6.9)	2 (8.7)	0 (0.0)

n, number; %, percentage; IV, intravenous; SC, subcutaneous; IFX, infliximab; VDZ, vedolizumab.

**Table 4 jcm-11-07296-t004:** Comparison of characteristics of patients treated with vedolizumab according to acceptance or refusal for switching.

	Total		Acceptance		Refusal		
	N= 36		N = 30 (83.3%)		N = 6 (16.7%)		
							*p* **
Gender, n (%)							0.2301
Male	16	(44.4)	12	(40.0)	4	(66.7)	
Female	20	(55.6)	18	(60.0)	2	(33.3)	
IBD, n (%)							0.1736
UC	21	(58.3)	19	(63.3)	2	(33.3)	
CD	15	(41.7)	11	(36.7)	4	(66.7)	
Previous surgery for IBD, n (%)							0.3464
No	29	(80.6)	25	(83.3)	4	(66.7)	
Yes	7	(19.4)	5	(16.7)	2	(33.3)	
Treatment dose escalation, n (%)							0.7642
No	20	(55.6)	17	(56.7)	3	(50.0)	
Yes	16	(44.4)	13	(43.3)	3	(50.0)	
Concomitant treatment, n (%)							1.0000
None	33	(91.7)	27	(90.0)	6	(100.0)	
5 ASA	2	(5.6)	2	(6.7)	0	(0.0)	
Methotrexate	1	(2.8)	1	(3.3)	0	(0.0)	
Median duration of treatment (year)		2.6		1.9		3.4	0.0150
Administration frequency, n (%)							0.1418
Every 12 weeks	1	(2.8)	0	(0.0)	1	(16.7)	
Every 8 weeks	19	(52.8)	17	(56.7)	2	(33.3)	
Every 6 weeks	4	(11.1)	4	(13.3)	0	(0.0)	
Every 4 weeks	12	(33.3)	9	(30.0)	3	(50.0)	

** Chi-2 test or Fisher’s exact test for qualitative variables, Wilcoxon test for quantitative variables. CD, Crohn’s disease; IBD, inflammatory bowel disease; UC, ulcerative colitis.

**Table 5 jcm-11-07296-t005:** Comparison of characteristics of patients treated with infliximab according to acceptance or refusal for switching.

	Total		Acceptance		Refusal		
	N= 94		N = 71 (75.5%)		N = 23 (24.5%)		
							*p* **
Gender, n (%)							0.9025
Male	48	(51.1)	36	(50.7)	12	(52.2)	
Female	46	(48.9)	35	(49.3)	11	(47.8)	
IBD, n (%)							0.8354
UC	23	(24.5)	17	(23.9)	6	(26.1)	
CD	71	(75.5)	54	(76.1)	17	(73.9)	
Previous surgery for IBD, n (%)							0.7342
No	64	(68.1)	49	(69.0)	15	(65.2)	
Yes	30	(31.9)	22	(31.0)	8	(34.8)	
Treatment dose escalation, n (%)							0.3410
No	41	(43.6)	29	(40.8)	12	(52.2)	
Yes	53	(56.4)	42	(59.2)	11	(47.8)	
Concomitant treatment, n (%)							0.8799
None	80	(85.1)	61	(85.9)	19	(82.6)	
5 ASA	8	(8.5)	6	(8.5)	2	(8.7)	
Azathioprine	3	(3.2)	2	(2.8)	1	(4.3)	
Methotrexate	3	(3.2)	2	(2.8)	1	(4.3)	
Median duration of treatment (year)		5.5		5.4		4.5	0.0008
Administration frequency, n (%)							0.5351
Every 10 weeks	8	(8.5)	5	(7.0)	3	(13.0)	
Every 8 weeks	60	(63.8)	44	(62.0)	16	(69.6)	
Every 6 weeks	16	(17.0)	14	(19.7)	2	(8.7)	
Every 4 weeks	10	(10.6)	8	(11.3)	2	(8.7)	

** Chi-2 test or Fisher’s exact test for qualitative variables, Wilcoxon test for quantitative variables. CD, Crohn’s disease; IBD, inflammatory bowel disease; UC, ulcerative colitis.

**Table 6 jcm-11-07296-t006:** Factors associated with patient’s acceptance for switching from IV to SC formulation.

Variables		SC Switch, n (%)	Bivariate OR (CI 95%)	*p*	Multivariate OR (CI 95%)	*p*
Gender						
	Male	48 (75.0)	1	0.4686		
	Female	53 (80.3)	1.4 (0.6–3.1)			
IBD type						
	UC	36 (81.8)	1	0.4206		
	CD	65 (75.6)	0.7 (0.3–1.7)			
Current treatment						
	Vedolizumab	30 (83.3)	1	0.3420		
	Infliximab	71 (75.5)	0.6 (0.2–1.7)			
Concomitant treatment						
	None	88 (77.9)	1	0.9683		
	5 ASA	8 (80.0)	1.1 (0.2–5.7)			
	Azathioprine	2 (66.7)	0.6 (0.0–6.5)			
	Methotrexate	3 (75.0)	0.9 (0.1–8.6)			
Treatment dose escalation						
	No	46 (75.4)	1	0.5572		
	Yes	55 (79.7)	1.3 (0.6–2.9)			
Administration frequency						
	Every 4 weeks	17 (77.3)	1	0.2704		
	Every 6 weeks	18 (90.0)	2.6 (0.5–15.5)			
	Every 8 weeks	61 (77.2)	1.0 (0.3–3.1)			
	Every 10 or 12 weeks	5 (55.6)	0.4 (0.1–1.9)			
Treatment duration at inclusion (month), median		101	0.9 (0.8–0.9)	0.0002	0.9 (0.8–0.9)	0.0002

5 ASA, 5-aminosalicylates; CI, confidence interval; IBD, inflammatory bowel disease; UC, ulcerative colitis; CD, Crohn’s disease; OR, odd ratio; SC, Subcutaneous; n, number.

## Data Availability

The data underlying this article are available in the article.

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
