# Peer review of "Inflammatory Bowel Disease Patients’ Acceptance for Switching from Intravenous Infliximab or Vedolizumab to Subcutaneous Formulation: The Nancy Experience"

_jcm, 2022, doi:10.3390/jcm11247296_

Round 1

Reviewer 1 Report

Well conduct and written study on the 2 new possible forms of Switch from IV to SC. 

Major comments: a need for a follow up on these patients after their decision to switch to answer the question of a real switch or just a survey with patient's agreement and duration of the switch, safety of it and pourcentage of switch back , as the publication answered only the very beginning of the process, and not real outcomes. 

A better understanding of the difference of patient population between the 2 biologics, which should / could be anlayzed separately as the acceptance rate, despite the fact that it is not significant , is clearly different.  In order to analyze and interpret these differences - further analyses should be performe and an additional table with patients' characteristics created and then either put as supplementary table or included in the paper.  

Some data which could play a role in the switch decision are missing: 

- How about patients with less stable disease ? fistulizing or stenotic disease ?

- Does the level of education of the patients and social situation influencing the decision ? 

--> could you consider this as well to complement your publication.

Minor aspects: Table 1 : "upper gastrointestinal", Table 2 : maybe add the accceptance rate per Medication Table 3. is one line missing or not correctly formatted ? 

Author Response

Reviewer: 1

Well conduct and written study on the 2 new possible forms of Switch from IV to SC.

A: We sincerely thank reviewer 1 for her/his feedback and this positive comment.

Major comments: a need for a follow up on these patients after their decision to switch to answer the question of a real switch or just a survey with patient's agreement and duration of the switch, safety of it and pourcentage of switch back , as the publication answered only the very beginning of the process, and not real outcomes.

A: We thank reviewer 1 for this valuable suggestion. These data were not available. We added this limit in the discussion.

A better understanding of the difference of patient population between the 2 biologics, which should / could be anlayzed separately as the acceptance rate, despite the fact that it is not significant , is clearly different.  In order to analyze and interpret these differences - further analyses should be performe and an additional table with patients' characteristics created and then either put as supplementary table or included in the paper. 

A: We thank reviewer 1 for this valuable suggestion. The manuscript was adapted accordingly.

Some data which could play a role in the switch decision are missing:

- How about patients with less stable disease ? fistulizing or stenotic disease ?

A: We thank reviewer 1 for this valuable suggestion. One of the inclusion criterion was clinical remission. There was no difference according to disease phenotype.

- Does the level of education of the patients and social situation influencing the decision ?

A: Unfortunately, these data was not available.

--> could you consider this as well to complement your publication.

Minor aspects: Table 1 : "upper gastrointestinal", Table 2 : maybe add the accceptance rate per Medication Table 3. is one line missing or not correctly formatted ?

A: We thank reviewer 1 for these valuable suggestions. The manuscript was adapted accordingly.

Reviewer 2 Report

The authors explore the inflammatory bowel disease patient’s acceptance for switching from intravenous (IV) infliximab or vedolizumab to subcutaneous (SC) as well as describe the causes of refusal or conversely the factors associated with acceptance. This article reports for the first time that a short duration of treatment was independently associated with a high switch acceptance rate. SC infliximab and vedolizumab hold potential for greater patient flexibility by self-administration, reducing travel and hospital attendance for infusion

The manuscript would be improved with minor clarifications:

·       Line 91: Remove parenthesis after “trough level”.

·       Table 1: There is an error in the percentage of Current Treatment: UC Remicade à 8/37 = 34.8 % (not 35.8). Please check also the percentages of CD (Biosimilar and Remicade): an approximation is needed to reach the 100% (52.2% for Biosimilar and 40.8 for Remicade).

As suggestion: could be useful for the reader define the IQR with both Q1 and Q3 values, in this way is more useful to appreciate the range. So please change IQR definition in all the manuscript.

Author Response

The authors explore the inflammatory bowel disease patient’s acceptance for switching from intravenous (IV) infliximab or vedolizumab to subcutaneous (SC) as well as describe the causes of refusal or conversely the factors associated with acceptance. This article reports for the first time that a short duration of treatment was independently associated with a high switch acceptance rate. SC infliximab and vedolizumab hold potential for greater patient flexibility by self-administration, reducing travel and hospital attendance for infusion

A: We sincerely thank reviewer 2 for her/his feedback and this positive comment.

The manuscript would be improved with minor clarifications:

  • Line 91: Remove parenthesis after “trough level”.

A: We thank reviewer 2 for this valuable suggestion. The manuscript was adapted accordingly.

  • Table 1: There is an error in the percentage of Current Treatment: UC Remicade à 8/37 = 34.8 % (not 35.8). Please check also the percentages of CD (Biosimilar and Remicade): an approximation is needed to reach the 100% (52.2% for Biosimilar and 40.8 for Remicade).

A: We thank reviewer 2 for this valuable suggestion. The manuscript was adapted accordingly.

As suggestion: could be useful for the reader define the IQR with both Q1 and Q3 values, in this way is more useful to appreciate the range. So please change IQR definition in all the manuscript.

A: We thank reviewer 2 for this valuable suggestion. After discussion with our methodologist, we would like to present the results with the IQR and not the Q1 and Q3 values.
